# Test-Time Layer Recurrence Enables Ultra-Deep Thinking in LLMs Without Chain-of-Thought

## Abstract

Transformers possess a **neural depth** of only $O(1)$, which restricts them to solving primarily **inductive** reasoning problems of bounded depth. In contrast, recurrent models allow the latent reasoning state $\mathbf{h}$ to be sequentially updated across arbitrarily many recurrent steps, enabling them to handle tasks that require deep reasoning. Owing to their non-recurrent architecture, Transformer-based large language models (LLMs) struggle on such tasks, performing poorly on domains like chess, multi-digit multiplication, and long-range counting. The emergence of Chain-of-Thought (CoT) reasoning has partially mitigated this limitation by simulating temporal recurrence through latent-to-text-to-latent conversion, thereby granting Transformer LLMs theoretically unbounded neural depth under ideal conditions. However, CoT comes at the cost of very long generation sequences and low time efficiency. Recent work has shown that reasoning depth can also be enhanced in the *vertical* direction by repeating Transformer layers, complementing the *temporal* depth introduced by CoT. These two approaches–horizontal depth extension via CoT and vertical depth extension via layer recurrence–exhibit distinct theoretical and practical properties, yet both hold strong promise for boosting the reasoning capabilities of Transformer-based LLMs. In this paper, we present both theoretical analysis and empirical comparasion pf these two paradigms, and demonstrate how each contributes to enhancing computational power and downstream performance, particularly in ultra-long reasoning scenarios where standard Transformers are most limited.

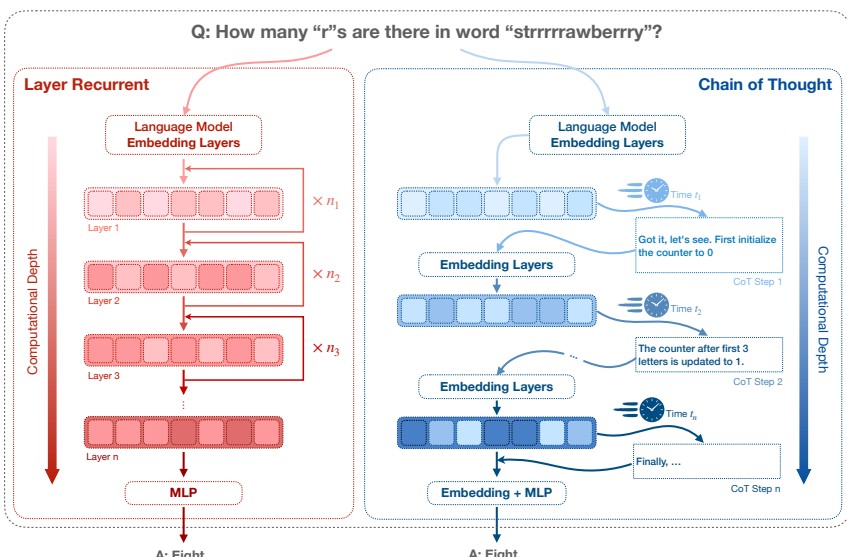

Figure 1: CoT increase computational depth by vector–string–vector conversion. Layer Recurrent LLM does so by increasing number of times each layer being used.

# 1 INTRODUCTION

The Transformer architecture (Vaswani, 2017) displaced recurrent models such as RNNs and LSTMs (Medsker et al., 2001), largely due to its optimization advantages: avoiding gradient vanishing and exploding issues and enabling efficient parallel training. However, the removal of recurrence also introduced a structural bottleneck. In practice, Transformers struggle with tasks requiring long-context reasoning, where their effective expressiveness is limited by finite context windows. While Transformers can theoretically simulate Turing machines (Pérez et al., 2021; Merrill & Sabharwal, 2023; Strobl et al., 2024), their actual computational power resembles classical space-bounded computation (Arora & Barak, 2009; Garrison, 2024). As a result, problems demanding deep sequential dependencies—such as parity checking—remain trivial for recurrent models (Delétang et al., 2022; Dziri et al., 2024) but challenging even for modern large language models (LLMs) (Achiam et al., 2023; Touvron et al., 2023; Jiang et al., 2023). The lack of true recurrence thus remains a fundamental limitation of the Transformer design.

To overcome this gap, two complementary test-time strategies have recently been explored (Figure 1): *Chain-of-Thought (CoT)* reasoning (Wei et al., 2022) and *Layer Recurrence* (Dehghani et al., 2018). These methods extend reasoning depth along orthogonal axes: CoT along the *horizontal* (time-wise) direction, and Layer Recurrence along the *vertical* (layer-wise) direction. CoT and its extensions (Nye et al., 2021; Kojima et al., 2022; Yao et al., 2024; Khot et al., 2022) create a surrogate recurrence loop: a latent state $\mathbf{h}_t$ is converted into text, which is then re-embedded into the model's latent space to continue computation. This latent → text → latent cycle enables temporal recurrence, effectively deepening reasoning through natural language. In contrast, Layer Recurrence directly reuses neural layers on the same latent state, iteratively refining $\mathbf{h}$ in continuous space. This vertical recurrence increases neural depth without the discretization bottleneck of CoT, providing a complementary mechanism for deeper reasoning.

Although both approaches can partially resolve the lack of true recurrence that Transformers struggle to learn (Zhang et al., 2024a), their theoretical limitations and practical effects have not been systematically compared. CoT provides a flexible mechanism for unbounded horizontal recurrence, but it incurs information loss through discretization and suffers from inefficiency as context grows (Long, 2023). Conversely, Layer Recurrence refines latent representations directly and is more time-efficient, yet it lacks the capacity for unbounded memory (as it does not extend the input sequence horizontally) and can collapse when recurrence is pushed too far without explicit training. These contrasting properties raise open questions regarding when each approach is preferable, how they complement one another, and to what extent they can reliably augment reasoning depth. In this work, we investigate these two paradigms both theoretically and empirically, highlighting their strengths, limitations, and implications for ultra-long reasoning in LLMs.

To concretely study the impact of these two approaches, we focus on the task of counting, which demands ultra-long reasoning when performed inductively and thus serves as a stringent test of reasoning depth. We apply both CoT-style reasoning and a layer-recurrent test-time approach with three distinct recurrence schemes: per-layer recurrence, whole-model recurrence, and interleaved block recurrence. By systematically varying the length of the counting sequence from 30 to 130, we thoroughly examine how each method scales with increasing reasoning demands and compare their effectiveness in sustaining long-horizon computation.

**Our investigation yields several key insights:** (1) both Layer Recurrence and Chain-of-Thought can be applied purely at inference time to augment the computational depth of large language models, a claim we validate both theoretically and empirically; (2) CoT provides unbounded external memory capacity while Layer Recurrence does not, but CoT suffers from information loss during the latent-to-token-to-latent conversion; (3) when computational depth must scale to ultra-long regimes, Layer Recurrence provides more stable augmentation, whereas CoT tends to degrade due to token conversion loss and attention limitations when reasoning spans thousands of tokens; (4) different implementations of Layer Recurrence yield broadly similar improvements, with differences arising mainly from which layers are repeated and how many times; (5) Over-recurrence of the total number of layers as well as choosing the task irrelevant layers to repeat can lead to direct degrading of the task performance, therefore layer choice matters; and (6) Layer Recurrence achieves greater time efficiency at test time compared to CoT, though excessively deep recurrence without training can result in unreadable outputs.

## 2 FIXED DEPTH OF TRANSFORMER AND WORKAROUNDS

### 2.1 FIXED DEPTH IN TRANSFORMER-BASED MODELS

The Transformer architecture is fundamentally a feedforward model of *constant depth* (Vaswani, 2017). Given an input sequence $x_{1:n}$, a Transformer with $L$ layers computes hidden states $\mathbf{h}_{1:n}^{(\ell)}$ for $\ell = 1, \ldots, L$ through successive applications of self-attention and MLP blocks:

$$\mathbf{h}^{(\ell)} = F^{(\ell)}(\mathbf{h}^{(\ell-1)}), \quad \ell = 1, \ldots, L, \tag{1}$$

where $\mathbf{h}^{(0)}$ is the embedding of $x_{1:n}$ and $F^{(\ell)}$ denotes the $\ell$-th Transformer layer. The final representation $\mathbf{h}^{(L)}$ is then mapped to output tokens via a linear projection and softmax.

Crucially, this computation has depth *bounded by* $L$, which is independent of the input length $n$. Unlike recurrent neural networks, where the hidden state $\mathbf{h}_t$ can be updated sequentially over $O(n)$ steps, the depth of a Transformer does not scale with $n$. As analyzed in Li et al. (2024); Zhang et al. (2024b); Sanford et al. (2024), this places Transformer models within the circuit complexity class $TC^0$ (Li et al., 2024; Feng et al., 2024), which consists of constant-depth threshold circuits. Models in $TC^0$ can implement certain parallelizable computations efficiently, but they cannot solve problems that require unbounded sequential depth, such as parity (Delétang et al., 2022; Dziri et al., 2024), majority with high thresholds, or iterated multiplication.

To illustrate, consider a function $f(x_{1:n})$ that requires reasoning depth proportional to $n$ (e.g., parity check). An RNN computes

$$\mathbf{h}_t = G(\mathbf{h}_{t-1}, x_t), \quad t = 1, \ldots, n, \tag{2}$$

and thus applies $O(n)$ sequential updates to $\mathbf{h}$. In contrast, the Transformer can only apply $O(L)$ transformations regardless of $n$, which yields a strict upper bound on its computational depth:

$$\text{Depth}_{\text{Transformer}}(n) = O(L), \quad \text{independent of } n. \tag{3}$$

It is important to emphasize that autoregressive generation in LLMs does not equate to recurrence. While autoregression unfolds tokens sequentially, each step discards the continuous latent reasoning trajectory $\mathbf{h}_t$ and replaces it with a discretized generated token (Zhang et al., 2024b). As a result, the computation is not a continuous recurrence over hidden states but rather a sequence of shallow, disconnected computations. This distinction explains why even extremely large LLMs behave no better than constant-depth threshold circuits when no additional mechanism for recurrence is introduced. The absence of continuous recurrence—and the corresponding growth of depth with sequence length—remains a fundamental bottleneck of Transformer architectures.

### 2.2 CoT MIMICS RECURRENCE FOR IMPROVED REASONING DEPTH

Chain-of-Thought (CoT) reasoning (Wei et al., 2022) provides a test-time mechanism for approximating recurrence in Transformer-based LLMs. The key insight, as shown by Li et al. (2024); Zhang et al. (2024b); Feng et al. (2024), is that although a Transformer cannot continuously update its latent state $\mathbf{h}$ over unbounded steps, it can instead *externalize* intermediate reasoning states into discrete tokens, and then re-embed these tokens to continue computation. This procedure simulates a recurrent update across time, extending reasoning depth beyond the fixed architectural bound and granting the model theoretical Turing Completeness under ideal conditions (Pérez et al., 2021; Merrill & Sabharwal, 2023; Strobl et al., 2024; Li et al., 2024).

Formally, let $\mathbf{h}_t \in \mathbb{R}^d$ denote the hidden state after $t$ reasoning steps. In a standard Transformer without CoT, $\mathbf{h}_t$ is mapped directly to the next token distribution $y_t = \text{Softmax}(W\mathbf{h}_t)$ and the generation process stops once an answer is produced. By contrast, in CoT we interpret the sampled token $o_t = \phi(\mathbf{h}_t)$ as an *intermediate thought*, where $\phi : \mathbb{R}^d \to V$ maps the dense hidden state to a discrete vocabulary $V$. A sequence of such tokens, $\{o_1, \ldots, o_k\}$, represents a discretized quantization of $\mathbf{h}_t$ into symbolic form. These tokens are then re-embedded back into latent space via $\psi : V \to \mathbb{R}^d$, giving $\tilde{\mathbf{h}}_{t+1} = \psi(o_t)$, and the Transformer continues with $\mathbf{h}_{t+1} = F(\tilde{\mathbf{h}}_{t+1}, \mathbf{h}_{\leq t})$.

> **Essence of Layer Recurrence**
>
> $$\mathbf{h}_t \xrightarrow{\phi} \{o_1, \ldots, o_k\} \xrightarrow{\psi} \mathbf{h}_{t+1}$$

This vector–string–vector loop should be contrasted with a true recurrent model (e.g., RNN), where the hidden state evolves continuously as $\mathbf{h}_{t+1} = G(\mathbf{h}_t, x_t)$. In CoT, recurrence is only *simulated*, since the dense reasoning state is repeatedly collapsed into discrete tokens before being restored. Nevertheless, each CoT step effectively adds one unit of reasoning depth, yielding an overall effective depth of $O(L \cdot T(n))$ for a base model of $L$ layers and a CoT sequence of length $T(n)$.

The strength of CoT lies in its flexibility: longer chains allow arbitrarily deep reasoning, and the intermediate tokens are interpretable as explicit reasoning traces. However, the fidelity of this simulated recurrence is limited by the lossy mapping $\phi$ and the finite vocabulary $V$. When $\mathbf{h}_t$ contains more information than can be serialized into $\{o_1, \ldots, o_k\}$, some computation is irretrievably lost, causing reasoning degradation, particularly in very long chains (Long, 2023). Thus, CoT offers an elegant but approximate mechanism for augmenting depth, in contrast to true recurrence where depth accumulates continuously without discretization.

## 2.3 Layer Recurrence Augments Depth Vertically

A complementary approach to extending reasoning depth in Transformers is *Layer Recurrence* (LR) (Dehghani et al., 2018), in which intermediate layers (or blocks of layers) are reapplied multiple times to the same hidden state. Unlike CoT, which simulates recurrence by discretizing $\mathbf{h}$ into tokens and re-embedding them, Layer Recurrence retains computation entirely in latent space, avoiding information loss due to quantization.

Formally, let $F^{(\ell)} : \mathbb{R}^d \to \mathbb{R}^d$ denote the transformation applied by the $\ell$-th Transformer layer, and $\mathbf{h}^{(\ell)}$ the corresponding hidden state. In a standard Transformer, the depth is fixed:

$$\mathbf{h}^{(\ell)} = F^{(\ell)}(\mathbf{h}^{(\ell-1)}), \quad \ell = 1, \ldots, L,$$

with overall depth $O(L)$. In Layer Recurrence, we allow repeated application of a subset of layers. For example, with recurrence factor $r$, we compute

> **Essence of Layer Recurrence**
>
> $$\mathbf{h}^{(\ell,k+1)} = F^{(\ell)}(\mathbf{h}^{(\ell,k)}), \quad k = 0, \ldots, r-1$$

where $\mathbf{h}^{(\ell,0)} = \mathbf{h}^{(\ell-1)}$ is the input to layer $\ell$. After $r$ repetitions, we set $\mathbf{h}^{(\ell)} = \mathbf{h}^{(\ell,r)}$ and continue with the next layer. More generally, recurrence can be applied to single layers (*per-layer recurrence*), groups of layers (*block recurrence*), or even the entire Transformer stack (*whole-model recurrence*).

The effective depth of such a model is thus $\text{Depth}_{\text{LR}} = O\left(\sum_{\ell=1}^{L} r_\ell\right)$ where $r_\ell$ is the recurrence factor applied to layer $\ell$. For uniform recurrence with $r_\ell = r$, this simplifies to $O(L \cdot r)$. In contrast to CoT, where depth grows with the number of generated reasoning tokens $T(n)$, LR increases depth deterministically through repeated internal computation.

An important distinction is that in LR the hidden state trajectory $\mathbf{h}^{(\ell,k)}$ evolves continuously within $\mathbb{R}^d$, without ever being collapsed into discrete tokens. This makes LR more time-efficient and less lossy, but also less interpretable: there are no externalized "thought" tokens, and reasoning is entirely implicit. Moreover, since LR applies the same parameters multiple times, excessively deep recurrence without architectural or training adjustments may lead to representation collapse or unstable dynamics. Nevertheless, under moderate recurrence factors, LR offers a principled way to vertically extend reasoning depth, complementing the horizontal extension provided by CoT.

## 2.4 Formal Comparison: CoT vs. Layer Recurrence

The two approaches differ fundamentally in how they augment reasoning depth and how they manage memory for computation.

**Depth complexity.** In Chain-of-Thought (Wei et al., 2022), each reasoning step corresponds to a vector–string–vector cycle $\mathbf{h}_t \mapsto \{o_1, \ldots, o_k\} \mapsto \mathbf{h}_{t+1}$, where $\phi : \mathbb{R}^d \to V^k$ discretizes the latent state into tokens and $\psi : V^k \to \mathbb{R}^d$ re-embeds them. The number of such conversions equals the CoT length $T(n)$, which determines how many times the hidden state can be recomputed. Thus the effective depth grows as $\text{Depth}_{\text{CoT}}(n) = O(L \cdot T(n))$, where $L$ is the base Transformer depth. Depth therefore scales adaptively with the number of reasoning steps.

In Layer Recurrence (Dehghani et al., 2018), depth is augmented by repeatedly applying layers. If the $\ell$-th layer is repeated $r_\ell$ times, then its contribution to depth is $r_\ell$. Summing across all layers gives $\text{Depth}_{\text{LR}} = O\big(\sum_{\ell=1}^{L} r_\ell\big)$. For uniform recurrence $r_\ell = r$, this reduces to $O(L \cdot r)$. Unlike CoT, depth is fixed by the recurrence factor chosen at inference time, not by token sequence length.

**Memory complexity.** Reasoning also requires sufficient memory capacity. In CoT, each generated token is stored in the sequence and can be retrieved via self-attention. Hence the available memory grows with the number of CoT steps, yielding $\text{Memory}_{\text{CoT}} = O(T(n))$. In contrast, Layer Recurrence never externalizes intermediate states; it only preserves the most recent hidden representation $\mathbf{h}^{(\ell, r_\ell)}$ while discarding prior ones. Thus its memory remains constant with respect to reasoning length, $\text{Memory}_{\text{LR}} = O(1)$.

In short, CoT extends depth horizontally and provides unbounded external memory at the cost of discretization overhead, whereas LR extends depth vertically within latent space, offering efficient but bounded recurrence. These trade-offs are summarized in Table 1.

In short, CoT extends depth horizontally and provides unbounded external memory at the cost of discretization overhead, whereas LR extends depth vertically within latent space, offering efficient but bounded recurrence. A plain Transformer without either mechanism remains constant in both depth and memory. These trade-offs are summarized in Table 1.

Table 1: Comparison of depth and memory complexity between plain Transformers, Chain-of-Thought, and Layer Recurrence.

| Method | Depth Complexity | Memory Complexity |
|---|:---:|:---:|
| Transformer (no CoT / LR) | $O(L)$ | $O(1)$ |
| Chain-of-Thought (CoT) | $O(L \cdot T(n))$ | $O(T(n))$ |
| Layer Recurrence (LR) | $O(\sum_{\ell=1}^{L} r_\ell)$ | $O(1)$ |

## 3 COUNTING TASK AND LAYER RECURRENCE SCHEMAS

### 3.1 COUNTING AS COMPUTABILITY TASK

To study reasoning depth, we adopt the task of *counting occurrences of a specific symbol in a sequence*. Formally, given an input string $x \in \Sigma^n$ over an alphabet $\Sigma$, the objective is to compute

$$f(x) = \#\{i \mid x_i = a\},$$

the number of times a designated symbol $a \in \Sigma$ appears in $x$.

This task is well-suited as a benchmark because it requires only *constant memory* but *linear depth*. A single counter suffices to store intermediate state, since the computation can be written inductively as $c_0 = 0$ and $c_{t+1} = c_t + \mathbb{1}[x_{t+1} = a]$, with $f(x) = c_n$. However, computing $c_n$ requires sequentially updating the counter across all $n$ positions, making the depth complexity $\Theta(n)$.

From the standpoint of formal language theory, the associated decision problem

$$L_{a,k} = \{x \in \Sigma^* \mid \#\{i \mid x_i = a\} = k\}$$

is not regular: it cannot be recognized by a finite automaton with bounded states, since the automaton would need to distinguish arbitrarily many counts of $a$. At the same time, $L_{a,k}$ does not require the full generative power of context-free languages. Instead, it lies in the class of *counter languages*, which are precisely those recognized by a one-counter automaton (equivalently, a pushdown automaton with a single stack symbol). Counter languages strictly contain the regular languages but are a strict subclass of the context-free languages.

This places counting in an intermediate complexity regime: it requires unbounded depth for sequential updates ($O(n)$), but only constant-size memory (the counter value). Such characteristics make counting a canonical probe task for depth limitations in Transformers. Plain Transformers, with constant computational depth $O(L)$, provably cannot compute such functions beyond trivial sequence lengths, whereas recurrent models or depth-augmented variants (CoT or LR) can in principle succeed.

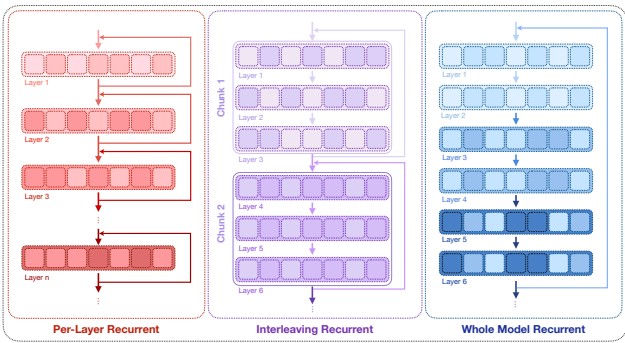

Figure 2: 3 mainstream paradigms of performing Layer Recurrent in LLMs.

## 3.2 THREE SCHEMAS OF LAYER RECURRENCE

Layer Recurrence (LR) augments depth by reapplying existing Transformer layers during inference. Formally, let $\{F^{(1)}, \ldots, F^{(L)}\}$ denote the $L$ layers of a Transformer. Standard forward propagation applies each layer once, yielding $\mathbf{h}^{(\ell)} = F^{(\ell)}(\mathbf{h}^{(\ell-1)})$. In LR, however, some or all of these transformations are repeated multiple times, increasing effective depth without modifying parameters. We study three natural recurrence schemas (see Algorithm 1–3 in Appendix and Figure 2): *per-layer recurrence*, where each layer $F^{(\ell)}$ is repeated $r$ times before moving on; *interleaved block recurrence*, where groups of layers are bundled into blocks and each block is repeated $r$ times; and *whole-model recurrence*, where the entire model $M = F^{(L)} \circ \cdots \circ F^{(1)}$ is reapplied $r$ times. All three achieve an effective depth of $O(\sum_\ell r_\ell)$, but differ in granularity and where recurrence is concentrated.

The effectiveness of these recurrence schemas may depend strongly on which layers are repeated. Prior research has suggested that Transformer layers are functionally specialized: early layers tend to capture surface-level lexical and syntactic features and perform token-to-latent conversion; middle layers are associated with semantic composition and reasoning; while late layers are more oriented toward organizing the latent state for fluent language generation. As a result, per-layer recurrence applied to middle layers could amplify reasoning ability, but recurrence of late layers may mostly affect fluency rather than reasoning, and excessive recurrence of early layers might introduce redundancy without deepening reasoning. Interleaved or whole-model recurrence, by contrast, distribute repetition across multiple functional roles, which may balance the trade-offs but also dilute the gains from targeting the most reasoning-relevant layers. Thus, the choice of recurrence schema implicitly shapes which cognitive abilities of the model are enhanced or impaired.

These three schemas all extend effective depth, but differ in granularity: per-layer recurrence repeats each layer individually, interleaving repeats groups of layers as blocks, and whole-model recurrence repeats the entire stack. As shown later, their empirical improvements are similar, with the main difference being *where* and *how many times* layers are repeated, rather than the recurrence structure itself.

## 4 EXPERIMENTAL

**Models.** We perform our analysis instruction-tuned large language model **Qwen2.5-72B-Instruct**(Team, 2024; Yang et al., 2024). A key reason for choosing Qwen2.5 is its distinctive architectural design: each Transformer block employs a combination of *pre-normalization* and *post-normalization* around attention and feedforward sublayers. This hybrid normalization scheme has been shown to stabilize representations both before and after transformation, ensuring that features

| Method | Recurrence | Length | Accuracy (% ↑) | Abs. Error (↓) | Valid (%) |
|--------|------------|--------|----------------|----------------|-----------|
| **Baseline** | – | 30 | 11.0 | 2.57 | 100 |
| LR (20–40) | $r = 1$ | 30 | 14.0 | 1.52 | 100 |
| CoT | – | 30 | **39.0** | **0.80** | 100 |
| **Baseline** | – | 45 | 2.0 | 4.52 | 100 |
| LR (20–40) | $r = 1$ | 45 | 4.0 | 3.96 | 100 |
| CoT | – | 45 | **28.0** | **2.45** | 100 |
| **Baseline** | – | 60 | 0.0 | 6.72 | 100 |
| LR (20–40) | $r = 1$ | 60 | 3.0 | 6.28 | 100 |
| CoT | – | 60 | **22.0** | **3.14** | 100 |
| **Baseline** | – | 80 | 0.0 | 12.42 | 100 |
| LR (20–40) | $r = 1$ | 80 | 3.0 | 6.96 | 100 |
| CoT | – | 80 | **9.0** | **5.58** | 100 |
| **Baseline** | – | 100 | 2.0 | 13.97 | 100 |
| LR (20–40) | $r = 1$ | 100 | **3.0** | **7.20** | 100 |
| CoT | – | 100 | 0.0 | 12.10 | 98 |
| **Baseline** | – | 130 | 1.0 | 14.42 | 100 |
| LR (20–40) | $r = 1$ | 130 | **3.0** | **9.40** | 100 |
| CoT | – | 130 | 0.0 | 16.61 | 89 |

Table 2: Counting performance on across different sequence lengths. Baseline Transformer (no CoT/LR), Layer Recurrence (per-layer recurrence on middle layers 20–40 with $r = 1$), and Chain-of-Thought (CoT) are compared. Metrics: accuracy (%, ↑ higher is better), absolute error shift (mean absolute deviation, ↓ lower is better), and valid output rate (%). Colored rows highlight CoT (blue) and LR (red).

are well regulated for subsequent computations and re-normalized before passing into the next layer. Such regulation is especially advantageous for *test-time Layer Recurrence*, where layers are reapplied multiple times without retraining. In this setting, stable normalization prevents feature explosion or drift across repeated applications, making Qwen2.5 particularly suitable for non-trained inference-time depth augmentation. Unless otherwise stated, decoding uses temperature 0, top_p = 1, and greedy output.

**Counting Task and Data Making.** We use the task of *symbolic-counting* task Delétang et al. (2022), which requires computational depth lienar to its length: given a sequence $x \in \{a, b\}^L$, the objective is to output $f(x) = \#\{i \leq L : x_i = a\}$. Sequences are drawn i.i.d. with $\Pr[x_i = a] = \Pr[x_i = b] = \frac{1}{2}$. We vary length $L \in \{30, 40, \ldots, 130\}$ and generate 100 independent instances per length (fixed seed; no duplicates within a length). This task requires *constant working memory* (a single counter) but *linear inductive depth* $\Theta(L)$, making it a stringent probe of depth augmentation.

**Metrics.** We set up *Chain-of-Thought* (CoT) and *Layer Recurrence* (LR) under matched and controlled-experiments conditions: (1)**CoT.** The model receives an instruction to **"reason step by step"** and must emit an explicit reasoning trace followed by a final numeric answer on the last line; (2) **Layer Recurrence (answer-only).** CoT is *disallowed*. The model must output a single numeric token as the final answer (no intermediate text) upon the given input prompt. We evaluate three LR schemas with recurrence factor $r \in \{2, 4, 8\}$: (i) *Per-layer*: each layer $F^{(\ell)}$ is applied $r$ times before proceeding; (ii) *Interleaved blocks*: a partition $\{\mathcal{B}_j\}$ of layers is repeated $r$ times blockwise; (iii) *Whole-model*: the full stack $M = F^{(L)} \circ \cdots \circ F^{(1)}$ is applied $r$ times. These choices isolate *vertical* depth without externalized tokens.

**Metrics.** We report two primary metrics for each sequence length $L$ and experimental setting: *Accuracy*, defined as Acc $= \frac{1}{100} \sum_{i=1}^{100} \mathbb{1}[\hat{y}_i = y_i]$, and *absolute error*, defined as AE $= \frac{1}{100} \sum_{i=1}^{100} |\hat{y}_i - y_i|$. For relatively short lengths, accuracy is informative; however, once $L > 80$, most LLMs (including even GPT-5) without external tools achieve nearly zero accuracy on the counting task. In this regime, absolute error provides a more meaningful estimate of how closely the predicted answer $\hat{y}$ approaches the ground truth $y$, reflecting the degree of numeric shift rather than binary correctness.

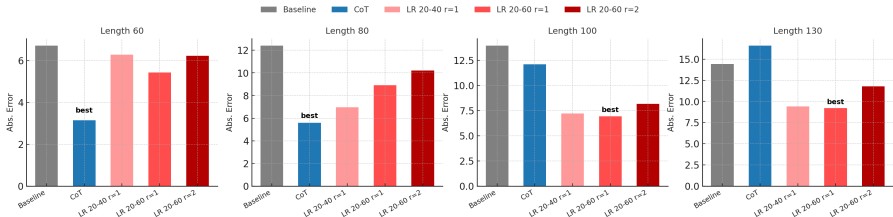

Figure 3: Absolute error of counting task across different sequence lengths ($L = 60, 80, 100, 130$) for five inference strategies: Baseline (gray), Chain-of-Thought (blue), and per-layer Layer Recurrence with 3 settings (shades of red). Error the lower the better here.

# 5 RESULTS ANALYSIS

## 5.1 BOTH CoT AND LR IMPROVE REASONING DEPTH.

The results in Table 2 highlight several key findings regarding the effectiveness of Chain-of-Thought (CoT) and Layer Recurrence (LR) in augmenting reasoning depth.

Across all sequence lengths, **both CoT and LR significantly outperform** the baseline LLM (no CoT/LR) in terms of accuracy and absolute error. At shorter lengths ($L = 30$–$60$), CoT achieves the lowest absolute error (e.g., $0.80$ at $L = 30$, compared to $2.57$ for the baseline and $1.52$ for LR), demonstrating its strength in simulating recurrent reasoning steps over moderately long contexts.

However, CoT degrades at long sequence lengths. As the sequence length increases ($L = 80$–$130$), the benefits of CoT diminish. Not only does the absolute error increase (e.g., $16.61$ at $L = 130$), but the valid output rate also drops (down to $89\%$). This instability arises because CoT relies on generating and attending over extremely long token sequences. The latent–token–latent conversion introduces compounding information loss, and global attention becomes unreliable across thousands of tokens, leading to degraded performance.

**LR dominates in ultra-long regimes.** In contrast, LR maintains stability and accuracy in ultra-long settings. For $L = 100$ and $L = 130$, LR significantly outperforms CoT: while CoT's absolute error rises to $12.10$–$16.61$, LR maintains a lower error ($7.20$–$9.40$) and a perfect $100\%$ valid output rate. This robustness stems from the fact that LR operates fully in latent space, avoiding discretization losses. Although its gains are modest at shorter lengths, LR becomes decisively superior in ultra-long contexts where CoT collapses.

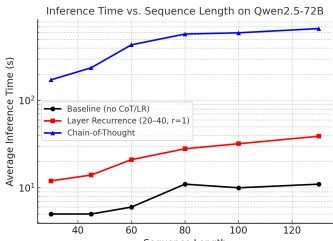

Figure 4: Average per-sequence processing time with each approach, no additional acceleration methods are used here.

**CoT and LR efficiency trad-offs.** Figure 4 shows that CoT, while effective for depth augmentation, suffers from prohibitively long inference times since each intermediate token must be generated sequentially, preventing parallelization and leading to runtimes above 600s at $L = 130$. In contrast, LR only reuses latent layers, adding a moderate linear overhead while remaining over an order of magnitude faster than CoT across all lengths. The baseline Transformer is fastest but fails to solve deep reasoning tasks, highlighting that LR provides the most favorable trade-off between depth and efficiency.

## 5.2 ANALYSIS OF LAYER RECURRENT STRATEGIES

We compare per-layer recurrence with interleaving recurrence schemes (Figure 5). Two main observations emerge. First, performance is primarily governed by the total number of *effective layers* applied. Regardless of whether recurrence is done sequentially (per-layer) or with interleaving gaps ($k = 2, 5, 10$), error rates improve as effective depth increases, and the curves track each other closely. Second, what matters in practice is which portion of the network is chosen for recurrence. Prior work suggests that lower layers are responsible for token-to-latent mapping, middle layers for

reasoning, and upper layers for output organization. All experiments here fix the same recurrence range (layers 20–60), which explains why differences between per-layer and interleaving strategies remain small. This indicates that while recurrence scheduling can vary, the dominant factors are (i) how much extra depth is applied, and (ii) which layer span is repeated, rather than the specific recurrence schema.

While layer recurrence effectively augments reasoning depth, we observe two major failure modes that can completely collapse model performance.

**Choosing the wrong layers.** Not all layers are equally suitable for recurrence. Prior work suggests that early layers are mainly responsible for token-to-embedding conversion, while late layers refine outputs into natural language. Repeating these layers destabilizes the latent trajectory: input embeddings drift, or output decoding loops collapse into gibberish. For example, applying whole-model recurrence (including both initial and final layers) fails to improve counting, and instead corrupts both representations and outputs.

**Over-recurrence.** Excessive recurrence can also destroy reasoning stability. As shown in the case study (Table 5.3), when the total effective depth becomes several times larger than the trained network (e.g., $8\times$), outputs degenerate into unreadable text. This likely arises from untrained wiring between repeatedly applied layers, causing embeddings to shift out of distribution. Moderate recurrence (e.g., $r = 1\text{--}2$) improves accuracy, but beyond a threshold the accumulation of instability outweighs any depth benefit.

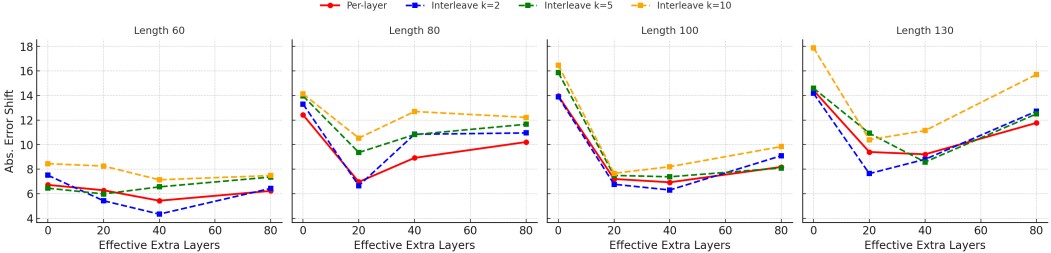

Figure 5: Effect of total effective layers on counting performance across per-layer and interleaving recurrence strategies.

## 5.3 Three schemas of Layer Recurrence

> **Case Study: Over-Recurrence Instability in Counting**
>
> **Example instance.** $x = $ a, b, a, a ,$\cdots$ b, a, b, (here $L = 100$, true count $y = 74$).
> **Observed outputs.**
> - $r = 1$ (no recurrence): 52                            (*incorrect*)
> - $r = 2$: 73              (*small bias, improved accuracy*)
> - $r = 4$: 120                          (*overcount*)
> - $r = 8$: th1rteen thirteen theeeee      (*gibberish / unstable*)

## 6 Conclusion

We studied two inference-time methods to deepen reasoning in LLMs: *CoT*, which adds unbounded but inefficient temporal depth, and *Layer Recurrence*, which adds efficient vertical depth with limited memory. Theory and experiments show CoT helps at moderate lengths, while LR is more stable and efficient for ultra-long reasoning. Together, they offer complementary paths toward deeper reasoning in Transformers.

## 7 REPRODUCIBILITY STATEMENT

To ensure the reproducibility of our findings, we have based our study on publicly accessible resources and detailed methodologies. The primary model used in our experiments, `Qwen2.5-72B-Instruct`, is publicly available. Our evaluation is centered on a synthetic symbolic counting task, for which the data generation process is fully described in Section 4, allowing for exact replication of the datasets. The implementation details for our core experimental conditions—Chain-of-Thought and the three Layer Recurrence schemas (per-layer, interleaved block, and whole-model)—are thoroughly outlined in Section 3 and Section 4. Key hyperparameters, including the recurrence factor r, specific layers targeted for recurrence, and decoding parameters (temperature, top-p), are explicitly stated. To further facilitate verification and extension of our work, we will release all source code and experiment scripts upon publication.

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

# A RECURRENCE VERSUS AUTOREGRESSION: A COMPUTATIONAL DEPTH ANALYSIS

At the core of a neural network's ability to perform sequential reasoning lies the mechanism by which it propagates information through time. In this section, we formally distinguish between two fundamental paradigms: true recurrence, which operates on latent states, and autoregression, which operates on generated outputs. We demonstrate that while both involve sequential processing, only true recurrence provides unbounded computational depth, and we frame Chain-of-Thought (CoT) (Wei et al., 2022) as an ingenious method for simulating the former using the machinery of the latter. Our analysis focuses on the architectural upper bounds on computability, abstracting away from the specifics of network optimization.

## A.1 RECURRENCE AND UNBOUNDED SEQUENTIAL COMPUTATION

A computational process is recurrent if a function iteratively operates on its own output. In a recurrent neural network (RNN) (Medsker et al., 2001), this is realized by feeding the hidden state $\mathbf{h}_{t-1}$ from the previous timestep back into the model to compute the new hidden state $\mathbf{h}_t$. For an input sequence $\mathbf{x}_n = (x_1, \ldots, x_n)$, the state transition at time $t$ is defined by the repeated application of a single parameterized function $g_\theta$:

$$\mathbf{h}_t = g_\theta(\mathbf{h}_{t-1}, x_t) \tag{4}$$

This architecture creates a dynamic system where the computational depth is directly proportional to the length of the input sequence, $n$. Since each application of $g_\theta$ represents a constant number of computational steps (determined by the network's layers), an RNN processing an input of length $n$ has a **computational depth of** $O(n)$. This property is what endows recurrent models with the native ability to handle tasks requiring arbitrarily deep sequential reasoning, such as parity checking or long-range counting. The hidden state $\mathbf{h}$ serves as a continuous, high-dimensional memory that is iteratively refined over time.

## A.2 THE CONSTANT-DEPTH LIMITATION OF AUTOREGRESSIVE TRANSFORMERS

The Transformer architecture (Vaswani, 2017) replaced temporal recurrence with a feed-forward, layer-wise structure. For an input sequence of length $n$, the hidden state $\mathbf{h}_t^{(i)}$ at position $t$ and layer $i$ is computed as a function of all hidden states from the previous layer, $\mathbf{h}_{1:t}^{(i-1)}$:

$$\mathbf{h}_t^{(i)} = g_\theta^{(i)}(\mathbf{h}_{1:t}^{(i-1)}) \tag{5}$$

Crucially, the computation flows sequentially through a fixed number of layers $m$, but not through time. There is no direct connection from $\mathbf{h}_{t-1}$ to $\mathbf{h}_t$ within the same layer. The final output state at position $t$, $\mathbf{h}_t^{(m)}$, is a function of the entire input prefix $\mathbf{x}_{1:t}$, not the previous hidden state $\mathbf{h}_{t-1}^{(m)}$:

$$\mathbf{h}_t^{(m)} = G_\theta(\mathbf{x}_{1:t}) \tag{6}$$

Consequently, the number of sequential computational steps is bounded by the number of layers $m$. For any given Transformer, this is a constant, resulting in a **computational depth of** $O(1)$.

This constant-depth nature is not altered by the model's **autoregressive** training or decoding procedure. In autoregression, the model generates an output token $o_t$ based on $\mathbf{h}_t$, and this token is then appended to the input sequence for the next step: $\mathbf{x}_{1:t+1} = (\mathbf{x}_{1:t}, o_t)$. This extends the input sequence but does not create a recurrent loop for the latent state. The computation for the next step must re-process the entire extended sequence from scratch. The discrete token $o_t$ is a low-dimensional projection of the rich, continuous state $\mathbf{h}_t$ and cannot serve as a substitute for it in subsequent high-depth computations. Therefore, autoregression alone does not increase the model's intrinsic reasoning depth.

## A.3 CHAIN-OF-THOUGHT: SIMULATING RECURRENCE VIA AUTOREGRESSION

Chain-of-Thought (CoT) prompting is a mechanism that ingeniously repurposes the autoregressive loop to simulate true recurrence. It bridges the gap by externalizing the computational state. The process can be decomposed into a 'latent → text → latent' cycle:

1. **State Externalization ($\mathbf{h} \rightarrow \mathbf{o}_{1:k}$):** At a given reasoning step, the model's internal hidden state $\mathbf{h}_t$ encodes the necessary information (e.g., the current board state in chess). CoT prompts the model to articulate this latent state as a sequence of natural language tokens $\mathbf{o}_{1:k}$ (a "thought").

2. **State Internalization ($\mathbf{o}_{1:k} \rightarrow \mathbf{h}'$):** This generated text is appended to the input context via the autoregressive mechanism. In the subsequent generation step, the model processes this text, re-embedding it into a new hidden state $\mathbf{h}'$ that approximates the original state $\mathbf{h}_t$.

This cycle effectively implements the recurrent update from Equation 4, where the text serves as the medium for transmitting the state $\mathbf{h}_{t-1}$ to the next step. If a CoT performs $T(n)$ such steps for an input of length $n$, it achieves a simulated **computational depth of** $O(T(n))$. This allows the Transformer to break free from its $O(1)$ architectural constraint and, under ideal conditions, approach Turing completeness (Pérez et al., 2021; Li et al., 2024).

However, this simulation is not without cost. The primary limitation is the **information bottleneck** inherent in the $\mathbf{h} \rightarrow \mathbf{o}_{1:k}$ conversion. The continuous, high-dimensional vector $\mathbf{h}$ must be discretized into a sequence of low-dimensional tokens, which is inevitably a lossy process. CoT variants like Tree-of-Thought (ToT) (Yao et al., 2024) and Graph-of-Thought (GoT) (Besta et al., 2024) can be viewed as attempts to mitigate this bottleneck. By generating multiple reasoning paths (ToT) or refining them iteratively (GoT), these methods aim to extract and preserve more of the essential information from the latent state $\mathbf{h}$ during the externalization step. Nonetheless, they still operate under the same fundamental mechanism and do not alter the overall depth complexity, which remains a function of the number of simulated recurrent steps.

### A.4 THE NECESSITY OF A UNIVERSAL CARRIER FOR SIMULATED RECURRENCE

The efficacy of CoT hinges on an implicit assumption: the model's output vocabulary is a **universal carrier**, capable of expressing any arbitrary computational state. Natural language is exceptionally well-suited for this role; it can describe chess boards, mathematical equations, data structures, and logical states with high fidelity. This universality is what allows an LLM to simulate recurrence across diverse domains.

This assumption breaks down for models trained on specialized, non-universal token sets. For example, a protein language model with a vocabulary of 20 amino acids (Lv et al., 2024) cannot use its autoregressive loop to externalize a complex reasoning state, as its tokens lack the required expressive power. Similarly, a chess model whose vocabulary is limited to move notation cannot generate a description of the board; it can only output the next action. In these cases, the autoregressive loop remains shallow, and the model's computational power is strictly confined by its $O(1)$ architectural depth. This highlights that the success of CoT is not just a property of autoregression but a synergistic effect of autoregression combined with a universally expressive output space.

## B ANALYSIS OF COMPUTATIONAL DEPTH AND MEMORY

The computational power of a neural network is fundamentally shaped by its architectural design, specifically its capacity for sequential processing (**depth**) and its requirements for storing intermediate state (**memory**). We analyze these properties for Multi-Layer Perceptrons (MLPs), Recurrent Neural Networks (RNNs), and Transformers, both with and without Chain-of-Thought (CoT) prompting.

### B.1 MLP: CONSTANT DEPTH AND MEMORY

A Multi-Layer Perceptron (MLP) (Popescu et al., 2009) is a feedforward network with a fixed number of layers, $m$. For any given input, the computation proceeds sequentially through these $m$ layers. Since $m$ is a constant that does not depend on the input size $n$, the computational depth of an MLP is $O(1)$. Similarly, the memory required is also constant. It only needs to store the fixed-size weight matrices and the activations for the current input. Therefore, an MLP operates with both $O(1)$ computational depth and $O(1)$ memory, making it unsuitable for tasks requiring reasoning over sequential data.

## B.2 RNN: SEQUENTIAL DEPTH WITH CONSTANT MEMORY

A Recurrent Neural Network (RNN) (Medsker et al., 2001) introduces a temporal loop, allowing it to process sequences of arbitrary length $n$. The model applies the same function $g_\theta$ at each timestep, using the previous hidden state $\mathbf{h}_{t-1}$ to compute the current one $\mathbf{h}_t$. This unrolls into a deep computational graph with a depth proportional to the sequence length, giving it a computational depth of $O(n)$.

However, a key architectural feature of an RNN is that it compresses the entire history of the input sequence into a single, fixed-size hidden state vector $\mathbf{h}_t$. Regardless of how long the input sequence $n$ is, the memory required to store the summary of the past is constant. This fixed-size bottleneck is what enables the RNN to process indefinitely long sequences with a small memory footprint, but it also makes it susceptible to losing information from the distant past. Thus, an RNN has $O(n)$ depth but only requires $O(1)$ memory to maintain its computational state.

## B.3 TRANSFORMER: CONSTANT DEPTH WITH LINEAR MEMORY

The Transformer architecture (Vaswani, 2017) abandons recurrence in favor of a parallelizable self-attention mechanism. Like an MLP, a Transformer has a fixed number of layers, $m$. Computation flows vertically through these layers, so its intrinsic sequential reasoning depth is limited by $m$, resulting in a constant depth of $O(1)$. This architectural choice is the primary reason standard Transformers struggle with tasks that require deep, iterative reasoning.

In terms of memory, the self-attention mechanism requires that for each token, the model has access to the key ($\mathbf{k}$) and value ($\mathbf{v}$) vectors of all previous tokens in the context. This necessitates a key-value (KV) cache that stores these vectors. As the input sequence length $n$ increases, the size of this cache grows linearly. Consequently, the Transformer has a memory complexity of $O(n)$. This trade-off is central to its design: it overcomes the RNN's information bottleneck by maintaining a complete history in memory, but at the cost of a memory footprint that scales with the context length.

## B.4 CoT: SIMULATED DEPTH WITH EXTENDED MEMORY

Chain-of-Thought (CoT) prompting (Wei et al., 2022) endows Transformers with a mechanism to simulate the temporal recurrence they lack. By generating intermediate reasoning steps as text, the model effectively externalizes its computational state $\mathbf{h}_t$ into a sequence of tokens. This latent $\rightarrow$ text $\rightarrow$ latent loop allows the model to perform a sequence of reasoning steps whose length, $T(n)$, depends on the problem's complexity, not the model's fixed architecture. This grants the model a simulated computational depth of $O(T(n))$.

This simulated depth comes at the cost of memory. The generated thoughts are appended to the input context, and the model must store the KV cache for this entire extended sequence. If the original input has length $n$ and the CoT has length $T(n)$, the total context length becomes $n + T(n)$. The memory complexity is therefore $O(n + T(n))$, which is dominated by the length of the reasoning chain and can be expressed as $O(T(n))$ for complex problems where $T(n) \gg n$.

Table 3: Comparison of computational depth and memory complexity for different neural architectures. Here, $n$ is the input sequence length and $T(n)$ is the length of the generated Chain-of-Thought.

| Model Architecture | Computational Depth | Memory Complexity |
|---|:---:|:---:|
| MLP | $O(1)$ | $O(1)$ |
| RNN | $O(n)$ | $O(1)$ |
| Transformer | $O(1)$ | $O(n)$ |
| Transformer + CoT | $O(T(n))$ | $O(T(n))$ |

## C THE ROLE OF MEMORY IN COMPUTATIONAL POWER

The computational capability of any system, from abstract machines to neural networks, is fundamentally determined by its memory architecture. Classical computability theory provides a formal

hierarchy of computational power—the Chomsky hierarchy—which is directly tied to the type and amount of memory a machine can access. By mapping neural architectures to this hierarchy, we can better understand their intrinsic strengths and limitations.

### C.1 FINITE MEMORY: THE FINITE STATE MACHINE AND THE RNN

The most basic computational model is the **Finite State Machine (FSM)**. An FSM's defining characteristic is its finite, constant-size memory, represented by its set of states. It can only remember which state it is currently in, regardless of the input's length. This $O(1)$ memory restricts FSMs to recognizing **regular languages**—patterns that can be identified without counting or tracking nested structures. For example, an FSM can recognize an alternating sequence of 0s and 1s, but it cannot verify if a string consists of $n$ 0s followed by $n$ 1s, as this would require counting, a task beyond its memory capacity.

The **Recurrent Neural Network (RNN)** is the neural analog of an FSM. Although an RNN processes sequences of arbitrary length $n$, giving it a computational depth of $O(n)$, it compresses the entire history of the sequence into a fixed-size hidden state vector $\mathbf{h}_t$. This hidden state, with its constant dimensionality, serves the same role as the finite states in an FSM. The RNN's memory is therefore $O(1)$. This architectural constraint explains why RNNs excel at capturing local sequential patterns but struggle with tasks requiring long-range dependencies, precise counting, or remembering nested information—the very tasks that are impossible for an FSM.

### C.2 STACK MEMORY: THE PUSHDOWN AUTOMATON

A step up in the hierarchy is the **Pushdown Automaton (PDA)**, which augments an FSM with an infinite stack. This stack provides a Last-In, First-Out (LIFO) memory structure. The ability to push and pop symbols from the stack allows a PDA to handle tasks that require unbounded memory with a specific structure, such as counting and matching nested pairs. Consequently, PDAs can recognize **context-free languages**, a class of problems that includes balancing parentheses or recognizing palindromes—tasks impossible for an FSM. The memory of a PDA is unbounded but constrained to a stack structure, allowing it to solve a broader, yet still limited, class of problems. While no direct mainstream neural architecture perfectly mirrors a PDA, models augmented with an explicit neural stack or those designed for recursive processing aim to capture similar capabilities.

### C.3 UNBOUNDED MEMORY: THE TURING MACHINE AND THE TRANSFORMER WITH COT

At the apex of this hierarchy lies the **Turing Machine**, the theoretical model for general-purpose computation. A Turing machine has access to an infinite tape that it can read from, write to, and move along in either direction. This tape serves as an unbounded, random-access memory. With this limitless memory, a Turing machine can execute any algorithm and recognize any **recursively enumerable language**, making it **Turing-complete**.

The **Transformer with Chain-of-Thought (CoT)** approximates a Turing machine. The Transformer's context window serves as the finite, but dynamically extendable, "tape." While a standard Transformer uses this context as read-only memory to process an input of length $n$, the CoT process transforms it into a read-write workspace. Each step in a CoT writes new information—the results of an intermediate computation—onto the tape (context window). The model can then read from this newly written information in subsequent steps to continue its computation. The length of the CoT, $T(n)$, represents the amount of tape used. Because this process can, in theory, continue indefinitely by extending the context, the memory becomes unbounded, scaling as $O(T(n))$. This ability to use its context as a read-write memory tape is what elevates the Transformer from a constant-depth processor to a system that can simulate the operations of a Turing machine, granting it the power to solve a far broader class of computational problems.

## D IMPLEMENTATION CODE

We implement and evaluate three distinct schemes for test-time Layer Recurrence, each differing in the granularity at which recurrence is applied: per-layer, whole-model, and interleaved block recurrence. The pseudocode for each approach is detailed in Algorithms 1, 2, and 3.

Table 4: The correspondence between computational models in the Chomsky hierarchy and neural network architectures, highlighting the central role of memory in defining computational power.

| Computational Model | Memory Type | Memory Complexity | Neural Network Analog | Problem Class |
|---|---|---|---|---|
| Finite State Machine | Finite States | $O(1)$ | RNN | Regular Patterns |
| Pushdown Automaton | Infinite Stack (LIFO) | $O(n)$ | Recursive/Stack-NNs | Nested/Context-Free |
| Turing Machine | Infinite Tape (R/W) | Unbounded | Transformer + CoT | General Computation |

**Per-Layer Recurrence** In this fine-grained approach (Algorithm 2), each of the $L$ Transformer layers is applied $r$ times to its input hidden state. The final output of the $r$-th repetition at layer $\ell$ serves as the input for layer $\ell + 1$. This method allows the iterative refinement process to occur at every step of the model's computational hierarchy.

**Whole-Model Recurrence** As the coarsest implementation (Algorithm 3), this scheme treats the entire Transformer model as a single composite function $M$. This function is applied $r$ times to the initial input embedding, with the output of one full pass through the model becoming the input for the next.

**Interleaved Block Recurrence** This scheme (Algorithm 1) offers a modular compromise between the other two. The Transformer's layers are partitioned into $m$ contiguous blocks. The algorithm iterates through these blocks, applying each block $r$ times before passing the resulting hidden state to the subsequent block. This allows for targeted depth extension within specific functional segments of the model architecture.

---

**Algorithm 1:** Interleaved Block Recurrence

---

**Input:** Input $x$, Transformer layers partitioned into blocks $\mathcal{B}_1, \ldots, \mathcal{B}_m$, recurrence factor $r$

$\mathbf{h}^{(0)} \leftarrow \text{Embed}(x)$;

**for** $j = 1$ **to** $m$ **do**
  **for** $k = 1$ **to** $r$ **do**
    **foreach** *layer* $F \in \mathcal{B}_j$ **do**
      $\mathbf{h} \leftarrow F(\mathbf{h})$;

**return** $\mathbf{h}$;

---

**Algorithm 2:** Per-Layer Recurrence

---

**Input:** Input $x$, Transformer layers $\{F^{(1)}, \ldots, F^{(L)}\}$, recurrence factor $r$

$\mathbf{h}^{(0)} \leftarrow \text{Embed}(x)$;

**for** $\ell = 1$ **to** $L$ **do**
  $\mathbf{h}^{(\ell,0)} \leftarrow \mathbf{h}^{(\ell-1)}$;
  **for** $k = 1$ **to** $r$ **do**
    $\mathbf{h}^{(\ell,k)} \leftarrow F^{(\ell)}(\mathbf{h}^{(\ell,k-1)})$;
  $\mathbf{h}^{(\ell)} \leftarrow \mathbf{h}^{(\ell,r)}$;

**return** $\mathbf{h}^{(L)}$;

---

# E INTUITION FOR TEST-TIME LAYER RECURRENCE: DECOUPLING PROCEDURAL LOGIC FROM DECLARATIVE KNOWLEDGE

The effectiveness of test-time Layer Recurrence (LR) without additional training can be understood by explicitly decoupling the *procedural logic* of reasoning from the *declarative storage* of knowledge within the Transformer architecture. Many algorithmic tasks—such as counting or iterative optimization—can be described as recurrent state-transition systems, where a state $s_t \in \mathcal{S}$ evolves

---

**Algorithm 3:** Whole-Model Recurrence

---

**Input:** Input $x$, Transformer model $M = F^{(L)} \circ \cdots \circ F^{(1)}$, recurrence factor $r$
$\mathbf{h}^{(0)} \leftarrow \text{Embed}(x)$;
**for** $k = 1$ **to** $r$ **do**
    $\lfloor \quad \mathbf{h}^{(k)} \leftarrow M(\mathbf{h}^{(k-1)})$;
**return** $\mathbf{h}^{(r)}$;

---

according to a fixed operator $g : \mathcal{S} \times \mathcal{X} \to \mathcal{S}$ given an input $x_t \in \mathcal{X}$:

$$s_{t+1} = g(s_t, x_t).$$

The overall computation is the $n$-fold composition $g^n$. We hypothesize that certain Transformer layers, particularly middle layers parameterized by $\theta_l$, implicitly learn high-dimensional embeddings of such universal operators. In this view, the layer transformation $F_{\theta_l} : \mathbb{R}^d \to \mathbb{R}^d$ approximates an isomorphic mapping of $g$, where the hidden state $h^{(l)}$ represents the abstract state $s_t$.

Applying LR with recurrence factor $r$ to the $l$-th layer then corresponds to the functional composition

$$(F_{\theta_l} \circ F_{\theta_l} \circ \cdots \circ F_{\theta_l})(h^{(l-1)}) = F_{\theta_l}^r(h^{(l-1)}).$$

This operation does not inject new logic but instead extends the computational trajectory of the *same* learned operator $F_{\theta_l}$. If $F_{\theta_l}$ encodes one step of a recursive process (e.g., incrementing a counter), its repeated application $F_{\theta_l}^r$ naturally unfolds this process for $r$ steps. In effect, LR deepens computation along a consistent logical path, refining the latent state $h$ through successive applications of the same transformation. Empirically, we find that for well-formed operators $F_{\theta_l}$, this iterative mapping remains stable for moderate $r$, avoiding chaotic dynamics or representational collapse. Thus, LR serves as a mechanism for amplifying a model's procedural capacity at inference time without modifying parameters $\theta_l$.

This stands in sharp contrast to knowledge-intensive tasks, such as open-domain question answering, which depend on a distributed, heterogeneous composition of functions across layers. Here, the final representation

$$h^{(L)} = (F_{\theta_L} \circ F_{\theta_{L-1}} \circ \cdots \circ F_{\theta_1})(h^{(0)})$$

is the outcome of a non-uniform pipeline, where each $F_{\theta_i}$ is specialized (e.g., for parsing, entity resolution, or fact retrieval). The parameters $\{\theta_1, \ldots, \theta_L\}$ jointly encode a structured knowledge base. In such cases, inserting LR at an intermediate layer $l$ computes

$$F_{\theta_L} \circ \cdots \circ F_{\theta_l}^r \circ \cdots \circ F_{\theta_1},$$

which disrupts the delicate functional pipeline. Repeating $F_{\theta_l}$ neither improves access to knowledge embedded in other layers nor preserves the alignment of intermediate representations, and often corrupts $h^{(l-1)}$ in ways detrimental to downstream processing.

In summary, the utility of LR is task-dependent: it excels in **iterative reasoning** problems, where computation aligns with a homogeneous recurrent operator, but yields diminishing or negative returns in **knowledge retrieval** problems, where reasoning relies on heterogeneous, multi-stage transformations.

## F  RELATED WORK

**The Horizontal Path: Simulating Recurrence with Chain-of-Thought.**  The Transformer architecture's core computational limitation is its fixed, constant-depth reasoning capability, a stark departure from the temporally deep processing of recurrent networks (Medsker et al., 2001; Delétang et al., 2022; Dziri et al., 2024). Because Transformers pass hidden states through a fixed number of layers rather than across time steps, they cannot natively perform the deep sequential computations required for many inductive reasoning tasks (Li et al., 2024; Zhang et al., 2024b). Chain-of-Thought (CoT) prompting (Wei et al., 2022) emerged as a landmark solution to this architectural bottleneck. By externalizing intermediate reasoning steps into the text context, CoT forces the model to simulate

the recurrence it otherwise lacks. This **horizontal** extension of reasoning effectively unrolls a computational trace into a text sequence, transforming the Transformer into a system that is theoretically Turing-complete (Pérez et al., 2021; Merrill & Sabharwal, 2023; Li et al., 2024).

However, this theoretical power is practically constrained by the model's finite context window, a limitation analogous to classical space-bounded computation theory (Arora & Barak, 2009; Garrison, 2024). The linear, unbounded growth of the reasoning trace makes vanilla CoT inefficient and ultimately infeasible for very long computations. This has motivated a family of structured reasoning methods that attempt to better manage this text-based computational space, including trees (Yao et al., 2024; Long, 2023), graphs (Besta et al., 2024), and various task decomposition strategies (Zhou et al., 2022; Khot et al., 2022; Sel et al., 2023). While these approaches refine the exploration of the solution space, they all operate within the same horizontal paradigm: overcoming the Transformer's depth limitation by externalizing computation into an explicit, ever-expanding textual scratchpad.

**The Vertical Path: Reintroducing Recurrence via Layer-Wise Depth.** An alternative and fundamentally different paradigm seeks to reintroduce recurrence directly into the Transformer's architecture, thereby extending its reasoning depth **vertically**. This approach is distinct from standard autoregression; while autoregression uses the previously generated discrete token $y_{t-1}$ to predict the next, true recurrence operates on the continuous, high-dimensional hidden state $h_{t-1}$, preserving far richer computational information (Zhang et al., 2024b). Foundational work like the Universal Transformer (Dehghani et al., 2018) introduced this concept of *layer recurrence*, where the same block of layers is applied iteratively to refine hidden state representations. This mechanism deepens computation within the latent space itself, offering a more efficient pathway to complex reasoning than the tokenization-re-embedding loop of CoT. While emulating true recursion—as seen in programming languages (Weiss et al., 2021; Liu et al., 2023) or term rewriting systems (Baader & Nipkow, 1998)—remains challenging for standard Transformers (Zhang et al., 2024a), layer recurrence presents a direct, architecturally-grounded solution.

**The Unexplored Frontier: A Systematic Comparison.** Despite both paradigms aiming to solve the same core problem—the Transformer's inherent lack of reasoning depth—the research on horizontal (CoT-based) and vertical (layer-recurrent) approaches has proceeded largely in parallel. The former offers flexibility and unbounded memory at the cost of efficiency and information loss through discretization, while the latter promises efficiency and lossless refinement within a bounded memory scope. Yet, their relative strengths, weaknesses, and fundamental trade-offs in the context of modern LLMs have not been systematically investigated. It remains an open question when one approach is preferable, how they might complement each other, and what their practical scaling limits are for tasks demanding ultra-long reasoning. Our work aims to fill this critical gap by providing the **first direct theoretical and empirical comparison** of these two orthogonal paradigms for augmenting the reasoning depth of Transformers.

# G  THE USE OF LARGE LANGUAGE MODELS (LLMs)

Large Language Models (LLMs) were employed as assistive tools in the preparation of this work. In particular, we used GPT-5 to support minor edits to academic writing, including drafting and refining sections. All scientific claims, methodological contributions, and experimental results were entirely conceived, implemented, and validated by the authors. The authors retain full responsibility for the content of this paper.

