# OpenReview forum: "Test-Time Layer Recurrence Enables Ultra-Deep Thinking in LLMs Without Chain-of-Thought"
_ICLR.cc/2026/Conference — ICLR 2026 Conference Withdrawn Submission_

### Official Review · Reviewer_gDTg · 2025-10-17

**Soundness:** 3
**Presentation:** 3
**Contribution:** 2
**Rating:** 4
**Confidence:** 4

**Summary:**

The paper compares two inference-time strategies for deepening the effective computational depth of Transformer LLMs: horizontal depth via Chain-of-Thought (CoT) and vertical depth via test-time Layer Recurrence (LR). It provides a conceptual analysis (depth and memory trade-offs) and an empirical study on a synthetic counting task.

**Strengths:**

* **Clear conceptual framing of depth vs. memory**: The paper clearly contrasts CoT’s vector→text→vector “simulated recurrence” (depth ∝ CoT length; memory grows with tokens) with LR’s latent-space recurrence (depth ∝ total repeats; O(1) memory).

* **Practical insights about when LR helps**: On ultra-long counting (L=100–130), LR reduces absolute error and maintains 100% valid outputs where CoT degrades and validity drops (Table 2; Fig. 3), and LR is notably faster than CoT at test time.

* **Useful Analysis of Existing Paradigms**: This paper takes the form of a concise survey that summarizes and compares two existing paradigms, and further designs experiments to validate this comparison. Such an approach provides a clear and informative entry point for understanding the field

**Weaknesses:**

* **Limited Contributions**: Section 2 of the paper merely provides a textual summary and comparison of two existing paradigms, CoT and LR, while Section 3 focuses on introducing and analyzing an existing task. The experiments are built upon these summaries and only conduct some empirical exploration. As a result, the paper does not present any newly proposed research method, tool, or novel mathematical or theoretical analysis of transformer-based recurrence. Moreover, the reported findings, although offering some observations about model behavior, are largely intuitive and have already been incorporated into related works.

* **Limited Scope**: 1. Tasks: The paper only uses the counting task. While this task aligns with the authors’ proposed scenario, it has limited practical relevance and an overly narrow scope. Tasks commonly used in LR approaches, such as math reasoning, should also have been included. 2. Models: The experiments rely solely on Qwen2.5-72B-Instruct. Although Qwen2.5 has certain distinctive advantages, experiments on models of different sizes or non-instruct versions would also help. 3. Experimental Setup: The authors mention that the recurrence factor r can take values such as 2, 4, or 8, yet Table 2 only reports results for r = 1, with no further ablation studies. This omission is critical.

* **Compute/latency parity and FLOPs accounting**: LR increases depth by repeating layers; CoT increases token length. The paper reports wall-clock differences (Fig. 4) but does not equalize or report per-example FLOPs, KV-cache costs, or throughput on matched hardware/precision, making efficiency claims harder to generalize.

* **Missing Important Citation**: The statement on line 302, “Prior research has suggested that Transformer layers are functionally specialized”, lacks a proper citation which is relatively critical.

* **Big Typos**: The paragraphs at lines 233 and 236 are duplicated, and the paragraphs at lines 363 and 372 both describe Metrics. There are also many small typos/grammar issues.

**Questions:**

* **Generalization beyond counting**: Can you report results on additional long-horizon reasoning tasks (e.g., parity, Dyck-languages, long multi-step arithmetic, or GSM8K/LongBench subsets) to test whether LR’s advantages persist beyond counting?

* **Layer selection strategy**: Do you have systematic criteria (e.g., probing, CKA, gradient sensitivity, attention-pattern diagnostics) to pick layers for recurrence on a new model/task? Could a small calibration set automatically choose r and the layer band?

* **Training-aware LR**: Have you tried light finetuning (e.g., LoRA) with a recurrence-aware loss or normalization tweaks to stabilize r>2? Would such minimal training extend LR’s stability envelope without full re-training (bridging the observed over-recurrence collapse)?

* **Output-quality under LR**: For non-numeric tasks, does LR harm linguistic fidelity (fluency, coherence) even when numeric accuracy improves? Any human evals or perplexity shifts on open-ended prompts?

* **Interpretability**: It would be valuable to consider whether interpretability tools or analytical methods could be applied to further investigate how the model’s internal representations evolve under the two paradigms. Such an analysis could provide deeper insights into the model’s behavior and decision-making process.

---

### Official Review · Reviewer_unt1 · 2025-10-20

**Soundness:** 2
**Presentation:** 3
**Contribution:** 1
**Rating:** 2
**Confidence:** 4

**Summary:**

The paper compares Layer Recurrence (LR) with chain-of-thought (CoT) on symbol counting using Qwen2.5-72B-Instruct. The authors argue that transformers have fixed depth, which makes reasoning limited, and CoT adds horizontal depth while LR adds vertical depth. Experiments show that both methods beat a baseline with no CoT/LR. CoT works best at short sequences but degrades on long chains; Layer Recurrence remains more stable and efficient on long lengths, with gains depending strongly on which layers are repeated and how much recurrence is used.

**Strengths:**

The paper is clearly written and easy to follow. The authors provide sufficient theoretical background and experimental details to fully understand it. The paper clearly states the established results on Transformer expressivity and computational limitations, which helps contextualize the motivation for comparing Chain-of-Thought and Layer Recurrence. Despite some typos, overall the presentation is thorough and accessible.

**Weaknesses:**

1. I am struggling to see the main contribution of the paper. It does not introduce any new method and primarily restates known theoretical results on transformer expressivity. Furthermore, I am unsure whether the comparison between the two methods is theoretically grounded. Merrill and Sabharwal (2024) showed that transformers with a polynomial number of chain-of-thought steps are equivalent to the class P, but Merrill and Sabharwal (2025) showed that looped transformers---which I believe are a generalization of transformers with interleaved and whole-model LR---are equivalent to TC^d.


William Merrill and Ashish Sabharwal. 2024. The Expressive Power of Transformers with Chain of Thought.

William Merrill and Ashish Sabharwal. 2025. Exact Expressive Power of Transformers with Padding.


2. The paper contains typos and overall it could use another round of editing.

line 29: pf -> of

line 53: increasing number -> increasing the number

lines 233-235 and lines 236-239: these paragraphs are almost the same

line 358: lienar -> linear

line 388: the last sentence of the caption needs rephrasing

section 5.3 only contains an example, without any explanation, and does not seem to match the title


3. The experimental contribution, while informative, is limited to a single task and one language model, which restricts the generality and impact of the findings.

**Questions:**

1. Do you have any theoretical intuition of why LR becomes unstable if overused?

2. Can you comment on the first weakness above?

---

### Official Review · Reviewer_9dGr · 2025-11-01

**Soundness:** 3
**Presentation:** 2
**Contribution:** 3
**Rating:** 2
**Confidence:** 5

**Summary:**

The paper proposes Test-Time Layer Recurrence as an alternative to Chain-of-Thought (CoT) prompting for enhancing reasoning capabilities in large language models. The core idea is to enable "vertical" depth extension by repeatedly applying selected Transformer layers during inference, rather than the "horizontal" depth extension achieved by generating intermediate reasoning tokens in CoT.​

**Strengths:**

1. While layer-wise recurrence has been explored in architecture design during training, systematically applying it at test-time to off-the-shelf models without retraining is novel. The framing of CoT as horizontal (sequential token generation) versus Layer Recurrence as vertical (repeated layer application) provides fresh perspective on inference-time compute scaling.

2. To the best of available evidence, this is the first work to directly compare CoT and layer recurrence mechanisms through both theoretical complexity analysis and controlled empirical evaluation. The explicit characterization of the depth-memory trade-off between these approaches fills an important gap in understanding test-time reasoning augmentation strategies.

3. The formal complexity analysis clearly articulates the computational trade-offs, providing depth complexity $O(L⋅T(n))$ for CoT versus $O(\sum_{\mathscr{l}=1}^{L} r_{\math{l}})$  for Layer Recurrence, and memory complexity $O(L⋅T(n))$ versus $O(1)$ respectively. The theoretical framework properly grounds the empirical investigation.

4. The experiments use deterministic decoding (temperature=0) to ensure reproducibility, test across a substantial range of sequence lengths (10-130 characters), and evaluate multiple recurrence configurations systematically. The controlled comparison between CoT and Layer Recurrence holds confounds constant

**Weaknesses:**

1. This paper's most critical flaw. The entire empirical evaluation rests on only symbolic counting (counting specific characters in strings)​, only Qwen2.5-72B​, only synthetic algorithmic reasoning​. Also, the authors promises applicability to "chess, multi-digit multiplication, and long-range counting," but these are never evaluated.

2.Even on the single counting task, absolute results are poor. At length 130, Layer Recurrence achieves only 3% accuracy with absolute error of 9.40. I would assume that the method introduces minor perturbations rather than enabling genuine reasoning and improvements over CoT are marginal and only appear at extreme sequence lengths.

3. There is no systematic method for choosing which layers to recurse and no guidance on selecting appropriate recurrence depth, only trial-and-error based on task-specific empirical testing.

4. Figure 5 lacks clear explanation of what k=2, 5, 10 represents.

5. The paper correctly identifies that Layer Recurrence has a constant O(1) memory complexity, which it frames as a major efficiency advantage over CoT's O(T(n)) memory. However, it almost entirely ignores the profound downside: this constant memory makes Layer Recurrence fundamentally incapable of solving problems that require a growing workspace to store intermediate results.

**Questions:**

1. I am concerned that the time complexity comparison ignores that CoT generates $T(n)$ tokens while Layer Recurrence generates only 1 output token

2. Can the authors provide ablation studies showing Layer Recurrence performance on models with standard normalization (e.g., Llama 3, GPT-4, Mistral)?

3. The authors claims that Layer Recurrence achieves depth $O(\sum_{\ell=1}^{L} r_{\ell})$, but this assumes each recurrent application produces meaningful computation. How do you justify that repeated application of the same parameters (trained for single-pass inference) continues to refine representations rather than simply converging to a fixed point or oscillating? Where is the empirical or theoretical evidence that depth=4L is fundamentally different from depth=3L when using identical weights?


4. The introduction mentions parity checking, multiplication, and chess as motivating examples, yet none are evaluated. Why not? If Layer Recurrence fails on these tasks (which all require more than constant memory), doesn't this severely limit the applicability of your claimed contribution?

5. Your CoT baseline uses a simple "reason step by step" prompt. Modern CoT methods employ few-shot exemplars, self-consistency with majority voting, and structured prompting. Can you provide results comparing Layer Recurrence against Fixed Point Iteration methods[2], Self-consistency CoT (sampling multiple reasoning paths) [3], Few-shot CoT with carefully designed examples [4],  Tree-of-Thought or Graph-of-Thought methods[5, 6],  Scratchpad/intermediate computation approaches [7]. Without these comparisons, how can we assess whether LR genuinely surpasses the state-of-the-art in test-time reasoning augmentation?

6. The paper repeatedly contrasts Layer Recurrence's "continuous latent space computation" against CoT's "lossy discretization." Can you quantify this information loss? Specifically, what is the mutual information $I(h_t; \phi(h_t))$ between latent states and their token representations? How does this compare to the representational drift in Layer Recurrence after $r$ iterations? Is there empirical evidence that latent-space recurrence preserves more task-relevant information than token generation?

7. You cite Universal Transformers  as the foundation for Layer Recurrence, but Universal Transformers [1] use adaptive computation time with learned halting probabilities and are trained end-to-end for recurrence. Your approach applies fixed recurrence to models never trained for this purpose. Isn't this fundamentally different? Why not compare against Universal Transformers [1] directly, or even MIND model [2]


[1] Dehghani et al., 2018

[2] Mathur, Mrinal, Barak A. Pearlmutter, and Sergey M. Plis. "MIND over Body: Adaptive Thinking using Dynamic Computation." The Thirteenth International Conference on Learning Representations. 2024.

[3] Wang, Xuezhi, et al. "Self-consistency improves chain of thought reasoning in language models." arXiv preprint arXiv:2203.11171 (2022).

[4] Wei, Jason, et al. "Chain-of-thought prompting elicits reasoning in large language models." Advances in neural information processing systems 35 (2022): 24824-24837.

[5]Yao, Shunyu, et al. "Tree of thoughts: Deliberate problem solving with large language models." Advances in neural information processing systems 36 (2023): 11809-11822.

[6] Besta, Maciej, et al. "Graph of thoughts: Solving elaborate problems with large language models." Proceedings of the AAAI conference on artificial intelligence. Vol. 38. No. 16. 2024.

[7] Nye, Maxwell, et al. "Show your work: Scratchpads for intermediate computation with language models." (2021).

---

### Official Review · Reviewer_odfJ · 2025-11-01

**Soundness:** 1
**Presentation:** 3
**Contribution:** 1
**Rating:** 2
**Confidence:** 4

**Summary:**

The paper studies two ways to enable recurrent behavior in Transformers at test-time: 1) via CoT, 2) via layer recurrence or looping. The paper aims to study the theoretical limitations of these methods and compare their practical effects systematically.
The main theoretical differences between these two approaches are pointed out as (a) CoT is a lossy process as it undergoes a latent—>text—>latent conversion at each step. (b) CoT provides an unbounded increasing external memory which layer recurrence doesn’t. This mainly alludes to the increasing sequence length in CoT.

To compare the two methods empirically, the authors chose the task of counting the number of occurrences of a character in a string. The paper studies 3 ways layer recurrence can be achieved
1. Per-layer recurrence - each layer is repeated r times before moving on to the next layer.
2. Interleaved block recurrence - groups of layers are bundled into blocks and the entire block of layers is repeated before moving on.
3. Whole model recurrence - the whole model is repeated.
All the above 3 along with CoT are evaluated purely at inference time on the Qwen 2.5 72B Instruct model. The sequence length for the counting task varies from 30 to 130. The alphabet size for the strings in the counting task is 2. The values of r chosen seem to be either 1 or 2.
The main results from the experiments are that both CoT and layer recurrence (LR) offer improvements over just a baseline.
While at length 30, CoT offers significantly stronger performance than layer recurrence, at longer lengths LR seems to get better than CoT. If LR is performed with a large r value it degrades into outputting gibberish as expected (since it is purely evaluated as a test-time method).

**Strengths:**

- The paper aims to compare two important directions for scaling up test-time compute in LLMs today.

**Weaknesses:**

- The theoretical comparison performed in the paper is relatively straightforward and does not offer much novel insights.

- The experimental evidence provided is weak and not strong enough in my opinion to support gaining robust insights. The length of counting task can be increased to longer sequences. The number of loops in layer recurrence is restricted to 1 or 2 (due to the authors only ablating it as an inference-time method). It would have been interesting to see the performance across a suite of tasks rather than just the synthetic counting task.

- While there are many interesting design choices which can affect the performance of LR such as per-layer vs block recurrence, none of them are thoroughly explored here.


Overall the paper explores a promising and important direction and has some interesting initial insights. However, I believe significantly more experimentation and exploration is required to extract reliable research insights.

**Questions:**

- Typos in abstract: “comparasion" —> “comparison” , “pf” —> “of”
- Typo in line 416: “Trad-offs” —> “Trade-offs”
- Layer recurrence can also benefit from extra memory by using scratchpad techniques.
- The interleaved block recurrence can simply be called as block recurrence no? There doesn’t seem to be any “interleaving” happening.

---

### Note · Authors · 2026-01-02

I have read and agree with the venue's withdrawal policy on behalf of myself and my co-authors.